# Morphological and Phylogenetic Characterization of Five Novel Nematode-Trapping Fungi (Orbiliomycetes) from Yunnan, China

**DOI:** 10.3390/jof9070735

**Published:** 2023-07-07

**Authors:** Fa Zhang, Yao-Quan Yang, Fa-Ping Zhou, Wen Xiao, Saranyaphat Boonmee, Xiao-Yan Yang

**Affiliations:** 1Center of Excellence in Fungal Research, Mae Fah Luang University, Chiang Rai 57100, Thailand; zhangf@eastern-himalaya.cn; 2Institute of Eastern-Himalaya Biodiversity Research, Dali University, Dali 671003, China; yangyaoquan0519@outlook.com (Y.-Q.Y.); 15721802339@163.com (F.-P.Z.); xiaow@eastern-himalaya.cn (W.X.); 3School of Science, Mae Fah Luang University, Chiang Rai 57100, Thailand; 4The Provincial Innovation Team of Biodiversity Conservation and Utility of the Three Parallel Rivers Region, Dali University, Dali 671003, China; 5Yunling Back-and-White Snub-Nosed Monkey Observation and Research Station of Yunnan Province, Dali 671003, China

**Keywords:** carnivorous fungi, new species, Orbiliaceae, phylogeny, trapping structure

## Abstract

Nematode-trapping fungi are widely studied due to their unique morphological structure, survival strategy, and potential value in the biological control of harmful nematodes. During the identification of carnivorous fungi preserved in our laboratory, five novel nematode-trapping fungi were established and placed in the genera *Arthrobotrys* and *Drehslerella* based on morphological and multigene (*ITS*, *TEF*, and *RPB2*) phylogenetic analyses. *A. hengjiangensis* sp. nov. and *A. weixiensis* sp. nov. are characterized by producing adhesive networks to catch nematodes. *Dr. pengdangensis* sp. nov., *Dr. tianchiensis* sp. nov., and *Dr. yunlongensis* sp. nov. are characterized by producing constricting rings. Morphological descriptions, illustrations, taxonomic notes, and phylogenetic analysis are provided for all new taxa; a key for *Drechslerella* species is listed; and some deficiencies in the taxonomy and evolution study of nematode-trapping fungi are also discussed herein.

## 1. Introduction

Nematode-trapping fungi (NTF) are a group of fungi that can produce unique structures (trapping structures) to capture nematodes [1,2,3]. They have attracted much attention for over 180 years since Corda (1839) reported the first species (*Arthrobotrys superba* Corda) because of their unique survival strategy, excellent application potential in nematode control, and significance of maintaining the balance of nematode populations in the ecosystem [4,5,6,7,8]. Orbiliomycetes NTF is the research focus of NTF due to their abundant species, diversified trapping structures, and mature research methods [3,9,10]. Currently, 119 Orbiliomycetes NTF species have been reported and divided into *Arthrobotrys*, *Dactylellina,* and *Drechslerella* based on their trapping structures according to modern molecular biology research [11,12,13,14].

*Arthrobotrys*, the most widespread and diverse (67 species) genus among Orbiliomycetes NTF, was established by Corda (1839) with *A. superba* Corda, which is characterized by 1-septate conidia growing in clusters on the nodes of the conidiophores [4]. With the improvement in the isolation method, more species were discovered, and the characteristic of *Arthrobotrys* was revised as producing obovoid, elliptic, pyriform 0–3-septate conidia on the nodes or short denticles of the conidiophores [2,15,16,17]. However, the taxonomy system based on these characteristics still needs to be clarified due to confusion caused by scholars attaching different importance to morphological features. The development of molecular biology has brought a significant breakthrough in the taxonomic study of NTF. Methods such as restriction fragment length polymorphism (RFLP), random amplified polymorphic DNA (RAPD), and molecular phylogenetics have gradually clarified the importance of the trapping structure in NTF classification [11,12,13,14]. Accordingly, the main characteristic of *Arthrobotrys* has also been revised again to produce adhesive networks to capture nematodes [3]. Species in *Arthrobotrys* are essential materials for developing bio-control agents for plant and animal parasitic nematodes because of their excellent competition, adaptation, and reproductive ability [18,19].

*Drechslerella* is the smallest genus (17 species) among Orbiliomycetes NTF, which separated from *Monacrosporium* by Subrammanian with *Dr. acrochaeta* (Drechsler) Subram as the type species based on conidia producing filamentous appendages at the apex [20]. However, Liu and Zhang (1994) pointed out that the filamentous appendage is not a stable and valid feature because it is formed by conidia germination, which is common in many *Arthrobotrys* and *Dactylellina* species [21]. Accordingly, *Drechslerella* is considered to be an invalid genus. Subsequently, the taxonomy of NTF was studied based on molecular phylogenetic analysis. All species produce constricting rings clustered into a monophyletic clade named *Drechslerella*, characterized by producing constricting rings composed of three cells and locking nematodes via the rapid expansion of the three cells [3,11,12,13,14]. This method of capturing nematodes mainly via mechanical force significantly differs from species in *Arthrobotrys* and *Dactylellina* (mainly capturing nematodes with adhesive material) [2,3]. Therefore, *Drechslerella* is a unique genus among Orbiliomycetes NTF and a key group in the origin and evolution study of carnivorous fungi.

NTF is a crucial node in fungal evolution and a good material for studying fungal adaptive evolution. The discovery of new species contributes to the development of related research and provides more materials for developing bio-control agents of parasitic nematodes. This research aims to report five new NTF species and list a key species of *Drechslerella* that has been studied less.

## 2. Materials and Methods

### 2.1. Sample Collection

Terrestrial soil and freshwater sediment samples involved in this study were collected from Yunnan Province, China. The detailed collection methods are the same as Zhang et al.’s [22].

### 2.2. Fungal Isolation

The soil sprinkling technique and baited plates method [3,23,24,25] were used to incubate nematode-trapping fungi (NTF) in the soil samples. The single-spore isolation method was used to obtain the pure culture of NTF. The details of the above three methods are the same as Zhang et al. [22].

### 2.3. Morphological Observation

The observation well and nematode baiting methods [26] were used to induce the trapping structure of NTF in accordance with Zhang et al. [22]. All micromorphological features, such as conidia, conidiophore, trapping structure, and chlamydospores, were photographed and measured with an Olympus BX53 differential interference microscope (Olympus Corporation, Tokyo, Japan).

### 2.4. DNA Extraction, PCR Amplification, and Sequencing

The total genomic DNA of isolates was extracted from the mycelium grown on potato dextrose agar (PDA) plates using a rapid fungal genomic DNA isolation kit (Sangon Biotech Company, Limited, Shanghai, China). The *ITS*, *TEF,* and *RPB2* regions were amplified with the primer pairs ITS4-ITS5 [27], 526F-1567R [28], and 6F-7R [29], respectively. The PCR amplification was performed according to Zhang et al. [22]. A DiaSpin PCR Product Purification Kit (Sangon Biotech Company, Limited, Shanghai, China) was used to purify the PCR products according to the user manual. The purified PCR products of the *ITS* and *RPB2* regions were sequenced in the forward and reverse directions using PCR primers, and *TEF* genes were sequenced using the primer pair 247F-609R [11] (BioSune Biotech Company, Limited, Shanghai, China).

Sequences were checked, edited, and assembled via SeqMan v. 7.0 [30]. The sequences generated in this study were deposited in the GenBank database at the National Center for Biotechnology Information (NCBI; https://www.ncbi.nlm.nih.gov/; accessed on 20 March 2023).

### 2.5. Phylogenetic Analysis

BLASTn search (BLASTn; https://blast.ncbi.nlm.nih.gov/; accessed on 11 March 2023) was used to compare the sequences generated in this study against the NCBI GenBank database. The BLASTn search results and the morphological features (trapping structure) of these five species indicated that they belong to the genus *Arthrobotrys* and *Drechslerella.* Therefore, all *Arthrobotrys* and *Drechslerella* taxa were searched in Species Fungorum (http://www.speciesfungorum.org; accessed on 12 March 2023) and checked individually according to the relevant documents to ensure that all *Arthrobotrys* and *Drechslerella* taxa were considered in this study [1,2,3,7,9,10,11,22,31,32,33]. All reliable *ITS*, *TEF,* and *RPB2* sequences of *Arthrobotrys* and *Drechslerella* taxa were downloaded from the GenBank database (Appendix A). Online program MAFFT v.7 (http://mafft.cbrc.jp/alignment/server/; accessed on 15 March 2023) [34] was used to generate the alignments of three genes, BioEdit v7.2.3 [35] was used to manually adjust the three alignments, and the three alignments were then linked with MEGA6.0 [36]. *Vermispora fusarina* YXJ 02-13-5 and *Vermispora leguminaceae* AS 6.0291 were set as outgroups. Phylogenetic trees were inferred via maximum likelihood (ML) and Bayesian inference (BI) analyses.

The best-fit optimal substitution models of *ITS*, *TEF*, and *RPB2* were selected as GTR+I+G, TrN+I+G, and GTR+I+G via jModelTest v2.1.10 [37] under the Akaike Information Criterion (AIC).

Maximum likelihood (ML) analysis was implemented using IQ-Tree v1.6.5 according to Zhang et al. [22]. The statistical bootstrap support values (BS) were computed using rapid bootstrapping with 1000 replicates [38].

Bayesian inference (BI) analysis was conducted with MrBayes v. 3.2.6 [39] according to Zhang et al. [22]. The remaining 75% of trees were used to calculate the posterior probabilities (PP) in the majority rule consensus tree.

FigTree v1.3.1 [40] was used to visualize the trees. The backbone tree was edited and reorganized using Microsoft PowerPoint (2013) and Adobe Photoshop CS6 software (Adobe Systems, San Jose, CA, USA).

## 3. Results

### 3.1. Phylogenetic Analysis

The combined *ITS*, *TEF*, and *RPB2* alignment dataset consisted of 104 *ITS* sequences, 60 *TEF* sequences, and 67 *RPB2* sequences from 66 *Arthrobotrys* taxa representing 62 valid species (plus our 2 new species), 32 *Drechslerella* taxa representing 21 valid species (plus our 3 new species), other related taxa in Orbiliaceae (*Dactylellina*: 4 species), and 2 outgroup taxa. The final dataset comprised 2038 characters (627 for *ITS*, 822 for *RPB2,* and 542 for *TEF*), among which 900 bp were constant, 1087 bp were variable, and 886 bp were parsimony informative.

A best-scoring maximum likelihood tree was performed with a final ML optimization likelihood value of −6158.611237. Within the Bayesian analysis (BI), the Bayesian posterior probabilities were evaluated with a final average standard deviation of the split frequency of 0.009264. The trees inferred by ML and BI showed slightly different topologies in some clusters, but both trees showed that all tested nematode-trapping fungi were clustered into two large clades, and five new species showed distinct divergence from known species. The best-scoring ML tree was selected to present herein (Figure 1), and the Bayesian majority rule consensus tree (BI) was also attached in the Appendix A (Appendix A).

The phylogenetic tree inferred from the *ITS*, *TEF,* and *RPB2* combined dataset placed five pairs of new isolates in *Arthrobotrys* and *Drechslerella*. *A. hengjiangensis* sp.nov. clustered with *A. jinpingensis* and *Orbilia jesu-laurae* with 99% MLBS and 0.98 BYPP support. *A. weixiensis* sp.nov. was sister to *A. globospora* with high support (99% MLBS, 1.00 BYPP). *Dr. pengdangensis* sp.nov. and *Dr. tianchiensis* sp.nov. were clustered together (89% MLBS). *Dr. yunlongensis* sp.nov. was clustered with four other species that produce fusiform conidia (100% MLBS, 1.00 BYPP) (Figure 1 and Appendix A).

### 3.2. Taxonomy

*Arthrobotrys hengjiangensis* F. Zhang & X.Y. Yang sp. nov. (Figure 2).

Index Fungorum number: IF900409; Facesoffungi number: FOF14151.

Etymology: The species name “hengjiangensis” refers to the name of sample collection site: Hengjiang County, Zhaotong City, Yunnan Province, China.

Material examined: CHINA, Yunnan Province, Zhaotong City, Hengjiang County, Hengjiang River, N 28°32′31.8″, E 104°19′09.5″, from freshwater sediment, 12 July 2014, F. Zhang. Holotype CGMCC 3.249834, preserved in the China General Microbiological Culture Collection Center. Ex-type culture DLUCC 34-1, preserved in the Dali University Culture Collection.

*Colonies* on PDA: initially white and turned to pale pink or yellowish after 2 weeks, cottony, growing rapidly, reaching 60 mm diameter after 10 days at 26 °C. *Mycelium*: partly superficial, partly immersed, composed of septate, branched, smooth, and hyaline. *Conidiophores*: 182.5–343 µm ($\bar{x}$ = 268.4 µm, *n* = 50) long, 3–5.5 µm ($\bar{x}$ = 3.7 µm, *n* = 50) wide at the base, gradually tapering upwards to the apex with 2.5–3.5 µm ($\bar{x}$ = 2.7 µm, *n* = 50) wide, erect, septate, unbranched or sometimes branched, producing a node at the apex or several separate nodes by repeated elongation of conidiophores, each node consisting of 3–8 papilliform bulges and bearing 3–8 conidia. *Conidia*: 14.5–29.5 × 9.5–18 µm ($\bar{x}$ = 19.9 × 12.7 μm, *n* = 50), obpyriform or drop-shaped, rounded at the apex, tapering towards narrow with tapering base, 0–2-septate (mostly 0 or 1-septate), and hyaline. *Chlamydospore* 7–14 × 5–10 µm ($\bar{x}$ = 9.5 ×7.6 μm, *n* = 50), cylindrical, globose or ellipsoidal, hyaline, and in chains when present. Nematodes were captured with adhesive networks.

Additional specimen examined: CHINA, Yunnan Province, Zhaotong City, Hengjiang County, Hengjiang River, N 28°32′31.8″, E 104°19′09.5″, from freshwater sediment, 12 July 2014, F. Zhang. Living culture XA190.

Notes: Phylogenetically, *Arthrobotrys hengjiangensis* clusters together with *A. jinpingensis* and *Orbilia jesu-laurae* with high support value (99% MLBS, 0.98 BYPP). *A. hengjiangensis* was 4.3% (27/626 bp) and 3.2% (20/620 bp) different from *A. jinpingensis* and *Orbilia jesu-laurae* in *ITS* sequences. Morphologically, these three species are similar in their conidia shape and the nodes of conidiophores [22,41]. However, *A. hengjiangensis* can be distinguished from *A. jinpingensis* by its wider conidia [*A. hengjiangensis*, 9.5–18 (12.7) µm versus *A. jinpingensis*, 6.5–14.5 (10.8) µm], 2-septate conidia with tapering base, and branched conidiophores [22]. The difference between *A. hengjiangensis* and *O. jesu-laurae* is that the conidiophores of *O. jesu-laurae* branched at the apex. In contrast, the conidiophores of *A. hengjiangensis* branched in the middle and upper parts. In addition, *Orbilia jesu-laurae* does not produce 2-septate conidia, while *A. hengjiangensis* does. Furthermore, the conidia produced by *A. hengjiangensis* have a more pointed base than those of O. *jesu-laurae*. The conidia of *O. jesu-laurae* are often slightly constricted at the septum, while those of *A. hengjiangensis* do not [41].

*Arthrobotrys weixiensis* F. Zhang & X.Y. Yang sp. nov. (Figure 3).

Index Fungorum number: IF900410; Facesoffungi number: FOF14152.

Etymology: The species name “weixiensis” refers to the name of sample collection site: Weixi County, Diqing City, Yunnan Province, China.

Material examined: CHINA, Yunnan Province, Diqing City, Weixi County, N 27°12′40.3″, E 99°05′24.2″, from terrestrial soil, 26 July 2014, F. Zhang. Holotype CGMCC3.24984, preserved in the China General Microbiological Culture Collection Center. Ex-type culture DLUCC 35-1, preserved in the Dali University Culture Collection.

*Colonies* on PDA: white, cottony, growing rapidly, reaching 55 mm diameter after 9 days in the incubator at 26 °C. *Mycelium*: partly superficial, partly immersed, composed of septate, branched, smooth, and hyaline. *Conidiophores* 165–364.5 µm ($\bar{x}$ = 253.4 µm, *n* = 50) long, 2.5–5 µm ($\bar{x}$ = 3.4 µm, *n* = 50) wide at the base, gradually tapering upwards to the apex 1.5–3 µm ($\bar{x}$ = 2.2 µm, *n* = 50) wide, erect, septate, unbranched, hyaline, producing 1–3 short denticles at the apex, and each denticle bearing a single conidium. *Conidia*: two types: I-type conidia: 22.5–39 × 14–27.5 µm ($\bar{x}$ = 27.8 × 17.7 μm, *n* = 50), drop-shaped or obovate, rounded at the apex, tapering towards narrow with subacute and truncate base, 1–2-septate (mostly 1-septate, usually located at the base), hyaline, with the largest cell located at the apex. II-type conidia: 30.5–48 × 14–27 µm ($\bar{x}$ = 36.7 × 19.5 μm, *n* = 50), fusiform, rounded at the apex, tapering towards narrow with subacute and truncate base, 1–2-septate (mostly 2-septate, usually located at both ends of the conidia), and hyaline, with the largest cell located at the middle of the conidia. *Chlamydospore*: 6–24 × 3.5–24 µm ($\bar{x}$ = 13.9 × 9.1 μm, *n* = 50), cylindrical, globose or ellipsoidal, hyaline or yellowish, and in chains when present. Nematodes were captured with adhesive networks.

Additional specimen examined: CHINA, Yunnan Province, Diqing City, Weixi County, N 27°12′40.3″, E 99°05′24.2″, from terrestrial soi, 26 July 2014, F. Zhang. Living culture FA675.

Notes: Phylogenetically, *Arthrobotrys weixiensis* forms a sister lineage to *A. globospora* (99% MLBS, 1.00 BYPP). There are 13.2% (64/484 bp) differences between them in *ITS*. Morphologically, the conidia shape of *A. weixiensis* and *A. globospora* are similar. They can be distinguished by their conidia size. The conidia of *A. weixiensis* are significantly larger than those of *A. globospora* [*A. weixiensis*, 30.5–48 (36.7) × 14–25 (19.5) µm versus *A. globospora*, 25–37.5 (30) × 15–22.5 (18) µm]. In addition, the conidiophore of *A. globospora* bears only a single conidium, while the conidiophore of *A. weixiensis* bears 1–3 conidia [2,3].

*Drechslerella pengdangensis* F. Zhang & X.Y. Yang sp. nov. (Figure 4).

Index Fungorum number: IF900411; Facesoffungi number: FOF14153.

Etymology: The species name “pengdangensis” refers to the name of sample collection site: Pengdang County, Nujiang City, Yunnan Province, China.

Material examined: CHINA, Yunnan Province, Nujiang City, Pengdang County, N 27°56′16.88″, E 98°39′8.71″, from terrestrial soil, 4 May 2018, F. Zhang. Holotype CGMCC 3.24985, preserved in the China General Microbiological Culture Collection Center. Ex-type culture DLUCC 37-1, preserved in the Dali University Culture Collection.

*Colonies* on PDA: white, cottony, growing slowly, reaching 40 mm diameter after 15 days in the incubator at 26 °C. *Mycelium*: partly superficial, partly immersed, composed of septate, branched, smooth, and hyaline. *Conidiophores*: 195.5–355 µm ($\bar{x}$ = 273.4 µm, *n* = 50) long, 2.5–5 µm ($\bar{x}$ = 3.5 µm, *n* = 50) wide at the base, gradually tapering upwards to the apex 2.5–4 µm ($\bar{x}$ = 2.4 µm, *n* = 50) wide, erect, septate, unbranched, and bearing a single conidium at the knob-like apex. *Conidia*: 30–45 × 17–27 µm ($\bar{x}$ = 38 × 22.4 μm, *n* = 50), ellipsoidal to subfusiform, rounded at the apex, tapering towards narrow with truncate at the base, 1–2-septate (mostly 2-septate), hyaline, with the largest cell located at the middle or apex of the conidia, where the base cell is tiny. *Chlamydospore*: not observed. Nematodes were captured with constricting rings; in the non-constricted state, the outer diameter is 19–28.5 µm ($\bar{x}$ = 24 µm, *n* = 50), and the inner diameter is 13–22.5 µm ($\bar{x}$ = 20.1 µm, *n* = 50).

Additional specimen examined: CHINA, Yunnan Province, Nujiang City, Pengdang County, N 27°56′16.88″, E 98°39′8.71″, from terrestrial soil, 4 May 2018, F. Zhang. Living culture DL53.

Notes: Phylogenetically, *Drechslerella pengdangensis* forms a sister lineage with another new species (*Drechslerella tianchiensis*) reported in this study, with 89% MLBS support. There are 15% (128/853 bp) differences in *ITS* sequence between them. Morphologically, *Dr. pengdangensis* can be easily distinguished from *Dr. tianchiensis* in the shape of the conidia and single conidiophore. *Dr. pengdangensis* is similar to *Dr. doedycoides* in their ellipsoidal to sub-fusiform conidia and simple conidiophore with knob-like apex [2,3]. However, *Dr. doedycoides* produces 3-septate conidia, while *Dr. pengdangensis* never. Moreover, the base cell of conidia produced by *Dr. pengdangensis* is significantly smaller than those of *Dr. Doedycoides* [2,3].

*Drechslerella tianchiensis* F. Zhang & X.Y. Yang sp. nov. (Figure 5).

Index Fungorum number: IF900412; Facesoffungi number: FOF14154.

Etymology: The species name “tianchiensis” refers to the name of sample collection site: Tianchi Nature Reserve, Yunlong County, Dali City, Yunnan Province, China.

Material examined: CHINA, Yunnan Province, Dali City, Yunlong County, Tianchi Nature Reserve, N 25°51′22.50″, E 99°13′38.43″, from burned forest soil, 28 May 2018, F. Zhang. Holotype CGMCC 3.24986, preserved in the China General Microbiological Culture Collection Center. Ex-type culture DLUCC 38-1, preserved in the Dali University Culture Collection.

*Colonies* on PDA white, cottony, growing slowly, reaching 40 mm diameter after 15 days in the incubator at 26 °C. *Mycelium* partly superficial, partly immersed, composed of septate, branched, smooth, hyaline. *Macroconidiophores* 186.5–305.5 µm ($\bar{x}$ = 248.1 µm, *n* = 50) long, 2.5–5 µm ($\bar{x}$ = 3.6 µm, *n* = 50) wide at the base, gradually tapering upwards to the apex with 1.5–3 µm ($\bar{x}$ = 2.2 µm, *n* = 50) wide, erect, septate, hyaline, unbranched or producing 1–2 short branches near the apex, each branch bearing a single conidium. *Microconidiophores* 137.5–245.5 µm ($\bar{x}$ = 183.7 µm, *n* = 50) long, 2–4 µm ($\bar{x}$ = 3.2 µm, *n* = 50) wide at the base, gradually tapering upwards to the apex with 1.5–3 µm ($\bar{x}$ = 1.8 µm, *n* = 50) wide, erect, septate, hyaline, unbranched, producing 3–12 short denticles near the apex, each denticles bearing a single conidium. *Conidia* two types: *Maroconidia* 30–41 × 14.5–24 µm ($\bar{x}$ = 36.2 × 18.7 μm, *n* = 50), ellipsoidal, rounded at the apex, tapering towards narrow with truncate base, 1–2-septate (mostly 2-septate), hyaline, with a largest cell located at the middle or apex of the conidia. *Miroconidia* 16–26.5 × 4.5–11.5 µm ($\bar{x}$ = 21.6 × 6 μm, *n* = 50), clavate or cylindrical, rounded at the apex, tapering towards narrow with truncate base, 0–1-septate (mostly 1-septate), hyaline. *Chlamydospore* not observed. Capturing nematodes with constricting rings, in the non-constricted state, the outer diameter is 20.5–27.5 µm ($\bar{x}$ = 24.7µm, *n* = 50), the inner diameter is 14.5–22 µm ($\bar{x}$ = 19.3µm, *n* = 50).

Additional specimen examined: CHINA, Yunnan Province, Dali City, Yunlong County, Tianchi Nature Reserve, N 25°51′22.50″, E 99°13′38.43″, from burned forest soil, 28 May 2018, F. Zhang. Living culture XJ353.

Notes: Phylogenetically, *Drechslerella tianchiensis* formed a sister lineage with *Dr. pengdangensis* (89% MLBS). Morphologically, *Dr. tianchiensis* is similar to *Dr. hainanensis* and the asexual morph of *Orbilia pseudopolybrocha* in their shape of macroconidia and microconidia. The difference between *Dr. tianchiensis* and *Orbilia pseudopolybrocha* is that the macro-conidiophore of the latter is simple and bears a single conidium, while some macro-conidiophore of *Dr. tianchiensis* produces 1–2 short branches near the apex and bears 1–2 conidia. The conidia of *Dr. tianchiensis* are significantly larger than those of *O. pseudopolybrocha* (*Dr. tianchiensis*, 30–41 (36.2) × 14.5–24 (18.7) µm versus *O. pseudopolybrocha*, 26–30 × 16–22.2 µm) [33]. *Dr. tianchiensis* can be easily distinguished from *Dr. hainanensis* by its 1–2-branch macro-conidiophore and wider microconidia (*Dr. tianchiensis*, 16–26.5 (21.6) × 4.5–11.5 (6) µm versus *Dr. hainanensis*, 18.2–22.8 × 4.2–5.3 µm) [32].

*Drechslerella yunlongensis* F. Zhang & X.Y. Yang sp. nov. (Figure 6).

Index Fungorum number: IF900413; Facesoffungi number: FOF14155.

Etymology: The species name “yunlongensis” refers to the name of sample collection site: Yunlong County, Dali City, Yunnan Province, China.

Material examined: CHINA, Yunnan Province, Dali City, Yunlong County, N 25°52′27.91″, E 99°22′19″, from terrestrial soil, 3 June 2018, F. Zhang. Holotype CGMCC 3.20946, preserved in the China General Microbiological Culture Collection Center. Ex-type culture DLUCC 39-1, preserved in the Dali University Culture Collection.

*Colonies* on PDA: white, cottony, growing slowly, reaching 45 mm diameter after 15 days in the incubator at 26 °C. *Mycelium*: partly superficial, partly immersed, composed of septate, branched, smooth, and hyaline. *Conidiophores*: 164–331 µm ($\bar{x}$ = 239.8 µm, *n* = 50) long, 2.5–5 µm ($\bar{x}$ = 3.3 µm, *n* = 50) wide at the base, gradually tapering upwards to the apex 1.5–3µm ($\bar{x}$ = 2.1 µm, *n* = 50) wide, erect, septate, unbranched, hyaline, bearing a single conidium at the apex. *Conidia*: 36–54 × 17–27 µm ($\bar{x}$ = 47 × 23.6 μm, *n* = 50), drop-shaped or fusiform, rounded at the apex, tapering towards narrow with truncate base, 1–4-septate (mostly 4-septate), hyaline, with the largest cell located at the apex or middle of the conidia. *Chlamydospore*: 5–14 × 5.5–10 µm ($\bar{x}$ = 8.7 ×7.1 μm, *n* = 50), cylindrical, globose or ellipsoidal, hyaline, and in chains when present. Nematodes were captured with constricting rings; in the non-constricted state, the outer diameter was 19.5–27 µm ($\bar{x}$ = 23.1 µm, *n* = 50), the inner diameter was 15–21.5 µm ($\bar{x}$ = 18.9µm, *n* = 50).

Additional specimen examined: CHINA, Yunnan Province, Dali City, Yunlong County, N 25°52′27.91″, E 99°22′19″, from terrestrial soil, 3 June 2018, F. Zhang. Living culture YL402.

Notes: The phylogenetic analysis clustered *Drechslerella yunlongensis* with the other four fusiform conidia-producing species (99% MLBS, 1.00 BYPP). *Dr. yunlongensis* was 9.8% (55/559 bp), 8.1% (40/496 bp), 9.1% (51/559 bp), and 7.9% (47/596 bp) different from *Dr. aphrobrocha*, *Dr. bembicodes*, *Dr. coelobrocha*, and *Dr. xiaguanensis* in *ITS*, respectively. Morphologically, *Dr. yunlongensis* is also similar to these four species. However, the conidia of *Dr. yunlongensis* are bigger than those of *Dr. bembicodes* and *Dr. xiaguanensis* (*Dr. yunlongensis*, 36–54 (47) × 17–27 (23.6) µm versus *Dr. bembicodes*, 36–43.2 (40) × 16.8–21.6 (20.5) µm versus *Dr. xiaguanensis*, 33–52 (42.5) × 9.5–28 (15.5) µm); moreover, *Dr. bembicodes* produces obovoid, 1-septate microconidia, while *Dr. yunlongensis* does not; the conidia of *Dr. xiaguanensis* are mostly 3-septate, while the conidia produced by *Dr. yunlongensis* are mostly 4-septate [2,3,42]. The difference between *Dr. yunlongensis* and *Dr. aphrobrocha* is that *Dr. aphrobrocha* produces mostly 3-septate conidia, while *Dr. yunlongensis* produces mostly 4-septate conidia; the conidia of *Dr. yunlongensis* are smaller than that of *Dr. aphrobrocha* due to its smaller apical cell (*Dr. yunlongensis*, 36–54 (47) × 17–27 (23.6) µm versus *Dr. aphrobrocha*, 40–57.5 (51) × 15.5–35 (24.6) µm) [2,3]. *Dr. yunlongensis* can be distinguished from *Dr. coelobrocha* by its wider conidia (*Dr. yunlongensis*, 17–27 (23.6) µm versus *Dr. coelobrocha*, 16.8–21.6 (19.8) µm), and shorter base and apical cells [2,3]. Furthermore, *Dr. yunlongensis* produces cylindrical or ellipsoidal chlamydospores, while none of the four closely related species produces chlamydospores [2,3,42].

### 3.3. Key to Known Species of Drechslerella

We do not update the species key of *Arthrobotrys* in this study because it has been updated in Zhang et al. [22], and no more new species have been reported except the two new species reported in this study. Super-cell in the species key refers to the cell in the conidia significantly larger than other cells.

1. Conidia without super-cell…………………………………...………………………………21. Conidia with a super-cell………………………………………………………….…….……52. Conidia 1–3-septate…………………………………………………………………….……...32. Conidia 0–1-septate…………………………………………………………………….…...…43. 5–10 conidia cluster arrangement on a cluster of short denticles (5–10) at the apex of conidiophore, conidia 28.5–39.0 × 6.0–8.5 µm, microconidia cylindrical……………………………………………………………………………..*O. tonghaiensis*3. Conidiophore produce 3–8 short denticles by repeated elongation, conidia are cylindrical, botuliform, 20–45 (30) × 5–12.5 (6) µm, and do not produce microconidia…………………………………………………………………………….....*Dr. brochapaga*4. Conidia digitiform are mostly curved, 1-septate, 35–51.5 (42.1) × 6.5–8 (7.5) µm, with 3–13 conidia capitate arrangement at the apex of conidiophore…………..*Dr. dactyleoids*4. Conidia are elongated and ellipsoidal, straight, 0-1-septate, 7.8–12.9 × 3.3–4.2 µm………………………………………………………………..………….*Dr. yunnanensis*5. Conidia are sub-fusiform to fusiform……………………………………………………..…65. Conidia are ellipsoidal, elongate ellipsoidal, subellipsoidal, or obovate……………….126. Conidia are 1–2-septate…………………………………………………….....*Dr. acrochaetum*6. Conidia are 1–5-septate………………………………………………………………….…....77. Conidia are 1–4-septate, mostly 3-septate……………………………………………….…..87. Conidia are 1–5-septate, mostly 4-septate…………………………………………………108. Conidia are smaller in size, 33–52 (42.5) × 9.5–28 (15.5) µm, swollen at both ends of cells………………….…………………………………………………..…..*Dr. xiaguanensis*8. Conidia are bigger, sometimes more than 52 µm in length and usually greater than 15 µm in width; the cells at both ends are not enlarged……………………………….…..99. Conidia are wider, 40–57.5 (51) × 15.5–35 (24.6) µm, 2–4-septate, and conidiophore occasionally bear two conidia……………………………………....….…….*Dr. aphrobrocha*9. Conidia are narrower, 42.5–62.5 (47) × 15–22.5 (16.9) µm, 1–4-septate, sub-fusiform, and conidiophore bear a single conidium……………………………………..…*Dr. inquisitor*10. Conidia are 3–4-septate, smaller in size, 36–43.2 (40) × 16.8–21.6 (20.5) µm, producing obovoid, 1-septate microconidia………………………………………..….*Dr. bembicodes*10. Conidia are bigger, do not produce microconidia………………………………………1111. Conidia are 1–4-septate, 36–54 (47) × 17–27 (23.6) µm, producing cylindrical, globose, or ellipsoidal chlamydospore…………………………………………....*Dr. yunlongensis*11. Conidia are 2–5-septate, 45.6–55.2 (49.5) × 16.8–21.6 (19.8) µm, both ends cells are slender, and do not produce chlamydospore……………………………..*Dr. coelobrocha*12. Conidia are obovate and 1-septate…………………………………………………….….1312. Conidia are ellipsoidal, elongate ellipsoidal, and 0–3-septate………………………....1413. Conidia are obovate, 29–43 (35) × 15–19 (16.8) µm, base cells are pyramidal, with 3–8 conidia capitate arrangement at the apex of conidiophore….....................Dr. anchonia13. Conidia are obovate or sub-ellipsoidal, 35 × 24 µm, single conidium bear at the apex of conidiophore……………………………………………………………..…...*Dr. polybrocha*14. Conidiophore is branched or bears more than 1 conidium………………………….…1514. Conidiophore is unbranched, bears a single conidium……………………………..…..1615. Conidiophore is unbranched or produces 1–2 short branches near the apex, each branch bearing a single conidium, with conidia 30–41 (36.2) × 14.5–24 (18.7) µm, 1–2-septate………………………………………………………………...……*Dr. tianchiensis*15. Conidiophore is unbranched, bearing a loose head consisting of 2–12 conidia, with conidia 32.5–45 (38.9) × 17.5–25 (21.4) µm, 1–2-septate……………………..….*Dr. effusa*16. Conidiophore produces a swollen, knob-like apex………………………………..……1716. Conidiophore produces a truncated, non-swelling apex……………………………....2017. Produces cylindrical, clavate, or bottle-shaped, 1-septate microconidia……………...1817. Does not produce microconidia…………………………………………………………..1918. Macroconidia are bigger, 17.5–45 (34) × 17.5–25 (20.4) µm, 1–2-septate, mostly 1-septate, and microconidia are bigger, 23–40 (31.3)× 5–8 (6.8) µm……………………………………………………………………….....…*Dr. heterospora*18. Macroconidia are smaller, 26–30 × 16–22.2 µm, 0–2-septate, mostly 2-septate, and microconidia smaller, 14.7–23 × 3.3–6 µm…………………….………...*O. pseudopolybrocha*19. Conidia are bigger, 30–45 (38) × 17–27 (22.4) µm, 1–2-septate, and basal cells are tiny…………………………………………………………………...…...…*Dr. pengdangensis*19. Conidia are 25–52.5 (33.2) × 12.5–29 (17.3) µm, and 1–3-septate………….*Dr. doedycoides*20. Conidia are elongated and ellipsoidal, 1–3-septate, mostly 3-septate, 34–56.5 × 12.5–16.5 µm, and do not produce microconidia………………………………*Dr. stenobrocha*20. Conidia are ellipsoidal, 0–2-septate, and produce clavate or bottle-shaped microconidia……………………………………………………………………………………...2121. Macroconidia are thinner, 20–49.5 (38.5) × 8.5–15 (12) µm, 1–2-septate, mostly 2-septate, and microconidia wider, 6.5–22 (15.5) × 3.5–7 (5) µm……….……*Dr. daliensis*21. Macroconidia are 32.5–43 × 17–25 µm, 0–2-septate, mostly 1 or 2-septate, and microconidia are 18.2–22.8 × 4.2–5.3 µm…………………………..…..…………*Dr. hainanensis*

## 4. Discussion

Both the phylogenetic analysis in this study and previous studies divided NTF into two main clades based on the mechanisms by which they catch nematodes (the genus *Drechslerella* produces constricting rings to capture nematodes with mechanical force, and the genera *Arthrobotrys* and *Dactylellina* catch nematodes with adhesive traps) [11,12,13,14]. These results again emphasized the significance of trapping structure for species division and evolution. Different from previous studies, this study failed to cluster *Dactylellina* species into a stable cluster, possibly due to insufficient DNA data. We believe that as more DNA data are used, we will find more morphological or physiological features that match phylogenetic studies.

The evolution of nematode-trapping fungi (NTF) is one crucial node to understanding the history of fungal evolution because of its unique morphological characteristics and survival strategy [2,3,4,5]. Currently, the main focus of the evolution research on NTF is the evolution of the trapping structure [9,11,43]. However, on the one hand, the phylogenetic clade of *Drechslerella* in this study showed that some species with similar conidia morphology cluster stably into one branch, such as species in clade I producing fusiform conidia and species in clade II producing ellipsoidal conidia (Figure 1). Moreover, in the whole NTF, species that produce the same trapping structure can easily be divided into different groups according to their conidia. For example, *Drechslerella* species can be divided into two groups according to the presence or nonpresence of super-cell in their conidia, and all *Arthrobotrys* species can be divided into three groups according to their conidia shape [2]. In addition to the law above, as the most critical reproductive structure in the asexual generation of fungi, conidia should have crucial evolutionary significance in theory. Based on the above, conidia may also be an essential evolutionary feature for NTF and an important basis for the NTF classification. Similarly, are other structures or physiological characteristics of NTF experiencing the same problems as conidia (which have important evolutionary or taxonomic significances but have been neglected)? In conclusion, the evolution of organisms is a process of interaction between organisms and the environment. The evolution of a single structure (trapping structure) cannot represent the evolution of the NTF species. The excessive focus on the evolution of a single structure while ignoring the characteristics of the whole species may lead to the mistake of the blind man feeling the elephant.

The compilation logic of the key of *Drechslerella* species is that species are first roughly classified by those features that can be used to identify species and are easily distinguishable, such as whether the conidia produce a super-cell or not, the shape of the conidia (fusiform, elliptical, cylindrical, digitate, etc.), whether the conidiophore is branched or not, and the number of conidia on the conidiophore. Then, species are further classified by those features that can be used for species identification but require further measurement and observation, such as the detailed feature of macroconidia (number and position of the septum and the size of the macroconidia). Finally, morphologically similar species are distinguished by those characteristics that are uncertain whether they can be used for species identification but are differences between different species, such as the presence and features of microconidia, the detailed feature of the apex of the conidiophore, and the features of the chlamydospore. Even identifying *Drechslerella* species requires those morphological features that are not known to be valid, so how difficult would it be to identify the more complex *Arthrobotrys* and *Dactylellina* species based on these features alone? Therefore, follow-up research needs to systematically study all potential morphological characteristics to find more reliable characteristics for species identification.

Most of the sexual generations of Orbiliomycetes nematode-trapping fungi are members of *Orbilia* [3]. However, due to the morphological conservation of the sexual generations, there exists a phenomenon wherein one sexual species corresponds to several morphologically different asexual species [44]. Additionally, with the implementation of the one fungus, one name policy [45,46], asexual NTFs need to use sexual names when discovering their sexual generation (*Orbilia* sp.). This results in these different asexual species sharing the same sexual species name [44], which further leads to confusion in the classification system and relevant data in some databases (such as Genebank, https://www.ncbi.nlm.nih.gov/nuccore/?term=Orbilia+auricolor (accessed on 3 April 2023)). For this reason, we suggest that when reporting a pair of sexual and asexual species, it is necessary to discuss the difference between the sexual generation and known sexual species and, more importantly, consider the distinction between the asexual generation and known asexual generation. The naming of this pair of sexual and asexual species should be carefully evaluated separately, giving sexual and asexual generations different species names if necessary.

## Figures and Tables

**Figure 1 jof-09-00735-f001:**
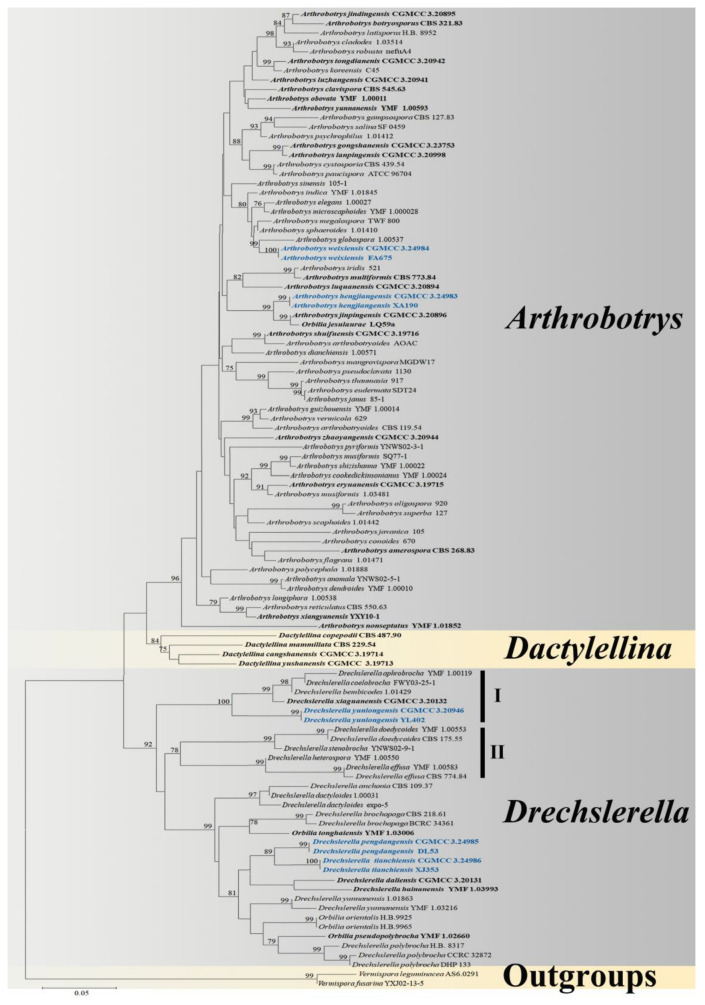
Maximum likelihood tree based on a combined *ITS*, *TEF* and *RPB2* sequence from 87 species of Orbiliaceae nematode-trapping fungi. Bootstrap support values equal to or greater than 70% are indicated above the nodes. The new isolates are in blue; type strains are in bold. The tree is rooted by *Vermispora fusarina* YXJ02-13-5 and *V. leguminacea* AS 6.0291.

**Figure 2 jof-09-00735-f002:**
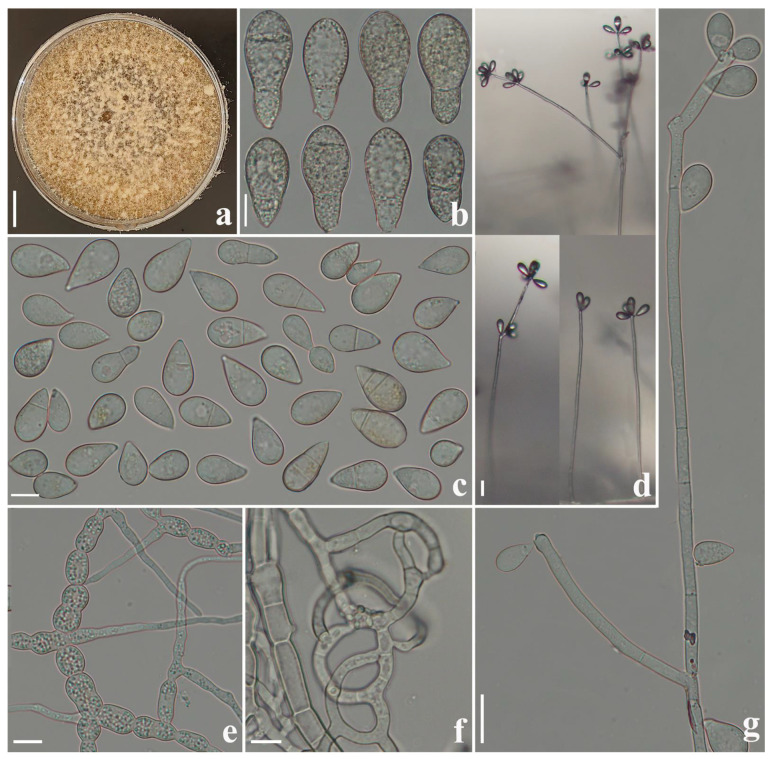
*Arthrobotrys hengjiangensis* (CGMCC 3.24983). (**a**) Colony. (**b**,**c**) Conidia. (**e**) Chlamydospores. (**f**) Trapping structure: adhesive networks. (**d**,**g**) Conidiophores. Scale bars: (**a**) = 1 cm, (**b**,**c**,**e**,**f**) = 10 µm, (**d**,**g**) = 20 µm.

**Figure 3 jof-09-00735-f003:**
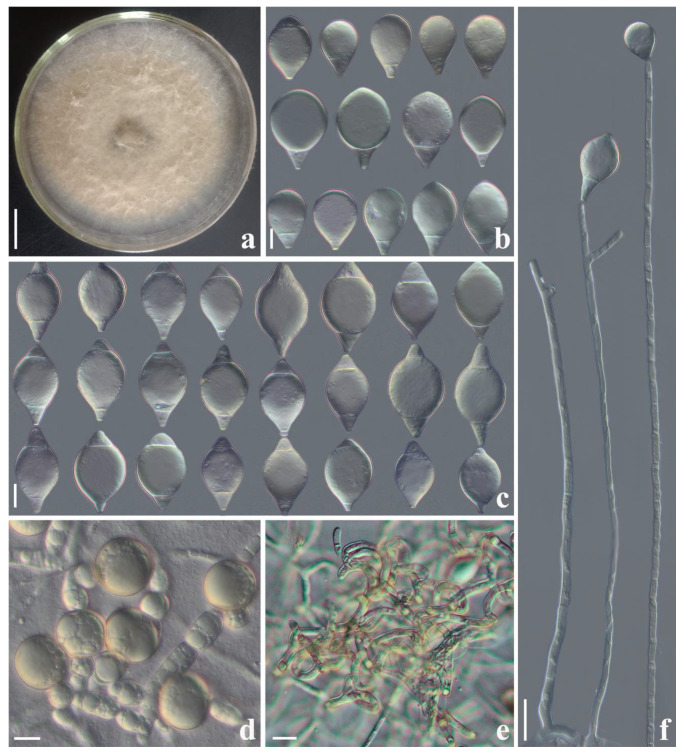
*Arthrobotrys weixiensis*. (CGMCC 3.24984). (**a**) Colony. (**b**,**c**) Conidia. (**d**) Chlamydospores. (**e**) Trapping structure: adhesive networks. (**f**) Conidiophores. Scale bars: (**a**) = 1 cm, (**b**–**e**) = 10 μm, (**f**) = 20 μm.

**Figure 4 jof-09-00735-f004:**
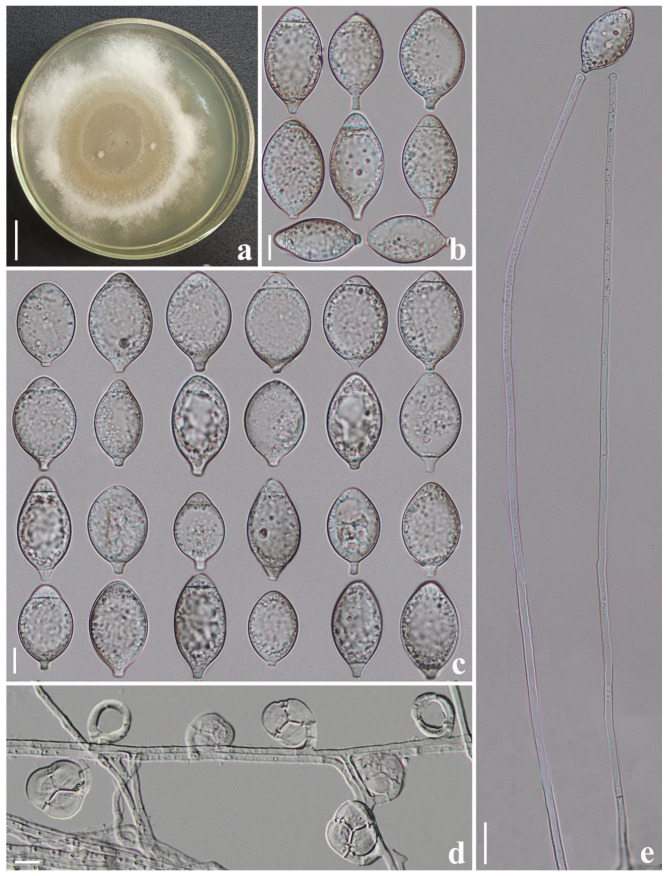
*Drechslerella pengdangensis* (CGMCC 3.24985). (**a**) Colony. (**b**,**c**) Conidia. (**d**) Trapping structure: constricting rings. (**e**) Conidiophores. Scale bars: (**a**) = 1 cm, (**b**–**d**) = 10 µm, (**e**) = 20 µm.

**Figure 5 jof-09-00735-f005:**
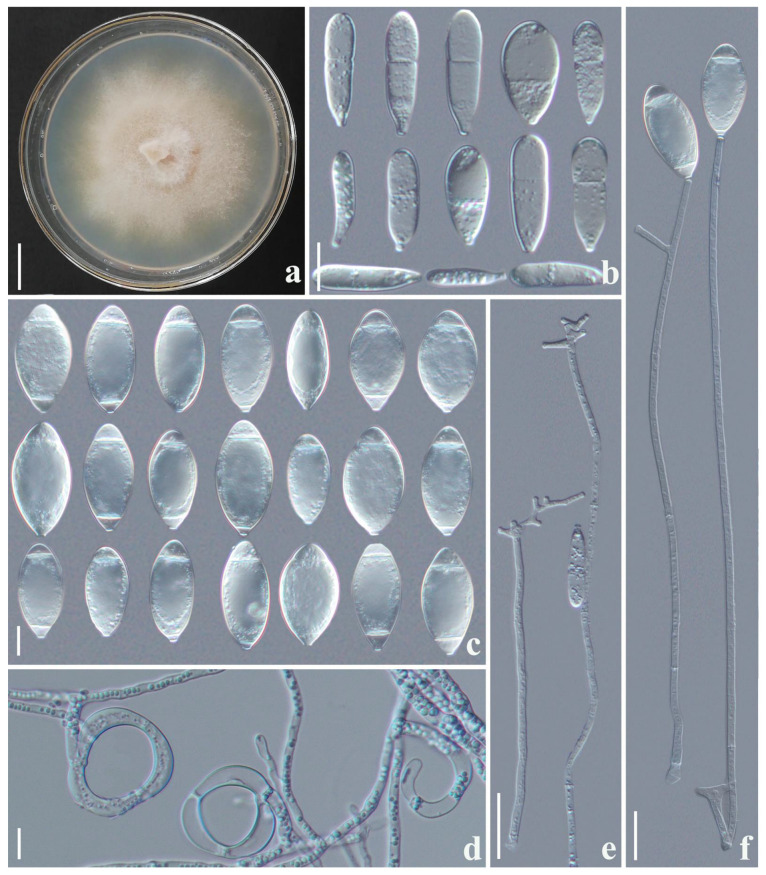
*Drechslerella tianchiensis* (CGMCC 3.24986). (**a**) Colony. (**b**) Microconidia. (**c**) Macroconidia. (**d**) Trapping structure: constricting rings. (**e**) Microconidiophores. (**f**) Macroconidiophores. Scale bars: (**a**) = 1 cm, (**b**–**d**) = 10 µm, (**e**,**f**) = 20 µm.

**Figure 6 jof-09-00735-f006:**
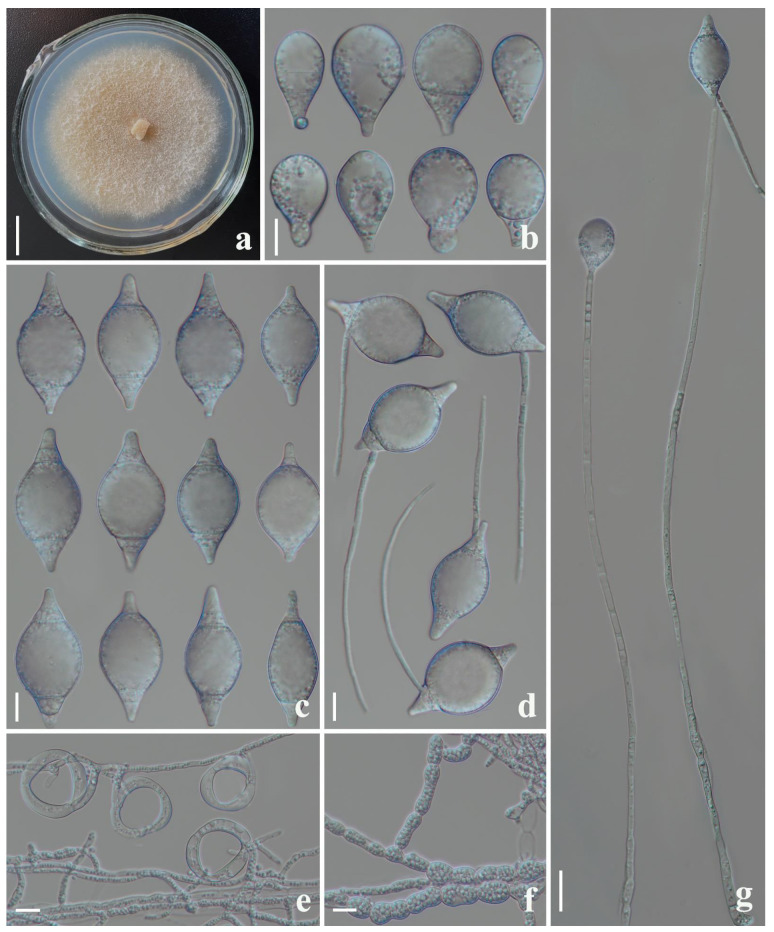
*Drechslerella yunlongensis* (CGMCC 3.20946). (**a**) Colony. (**b**,**c**) Conidia. (**d**) Germinating conidia. (**e**) Trapping structure: constricting rings. (**f**) Chlamydospores. (**g**) Conidiophores. Scale bars: (**a**) = 1 cm, (**b**–**d**) = 10 µm, (**e**,**f**) = 20 µm.

## Data Availability

The data that support the finding of this study are contained within the article.

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
