# Peer review of "Morphological and Phylogenetic Characterization of Five Novel Nematode-Trapping Fungi (Orbiliomycetes) from Yunnan, China"

_jof, 2023, doi:10.3390/jof9070735_

Round 1

Reviewer 1 Report

Dear Authors, i dont have something for your ms, it is excellent. I will add the an experiment with the pathogenity of this fungus at the nematodes. Except that i personally believe that your ms is good, very good. Bravo

Author Response

Dear reviewer

Thank you very much for rewiewing our manuscript and for your recognition of this study. I have double-checked the manuscript and made some appropriate corrections.

Thanks again for your help!

Best regards!

Reviewer 2 Report

A substantial and generally well done work. Excellent taxonomy part. However, some  weaknesses in the phylogenetics section and the data presentation:

Authors should consider to present the pages long Table 1 as Suppl Table.

The authors state that “the trees inferred by ML and BI showed slightly different topologies. (…) the best-scoring ML tree was selected for presentation (Figure 1).” If authors already do the work to analyze their data by two complementary methods, it might be  worth showing both outcomes, i.e. phylogenies, - and if the BI tree were only presented as Suppl Figure.

It is, by the way, not sound to indicate Bayesian posterior probabilities on the branches of the ML tree; please indicate these values in the BI tree.

Authors state that “fungi were clustered into three large clades”:

MLBS and BPP values at the root of the Arthrobotrys and Drechslerella clades (96%/0.94 and 92%/0.94) sufficiently support these clades. This is much less obvious for the Dactylellina clade receiving only 84% MLBS. There is no BPP indicated on this branch. Moreover, the branch rooting a common Arthrobotrys/ Dactylellina clade does not carry any statistical support indications: why? Both MLBS and BPP were below the rspective thresholds? Authors should discuss this outcome and what might be seen as a discrepancy with their interpretation.

Authors state that "The phylogenetic tree inferred from the ITS, TEF and RPB2 combined dataset placed five new species in Arthrobotrys and Drechslerella." Subsequently, MLBS and BPP values on the respective branches are discussed. Two respective remarks:

The study presents genetic data from 5 pairs of isolates. Both members of each pair according to descriptions stem from the same geographic origin. These sequences cluster in a pairwise manner with close to 100% MLBS / 1.00 BPP and with (apparently) zero terminal branch length and might therefore represent or 5 geographic clusters or even redundancies of isolation. There is no insight from representing each "new species" by two sequences that are not independent from each other. This appears redundancy of data.

In the paragraph following the above statement (lines 159-165) the study indicates varying MLBS and BPP values of the clustering of these new sequences with neighboring species. It might be more clearly indicated in the paragraph's argument that these values support assignment of the new fungal isolates at the genus level (i.e., in the manuscript's wording that "five (pairs of) isolates were placed in Arthrobotrys and Drechslerella").

However, the respective values and data can - by their very nature - not lend support to the placement of five new species to these genera. It should be pointed oout at the end of this paragraph that the arguments for species introduction are developed one-by-one and mostly on morphological rather than phylogenetic grounds in the subsequent taxonomy section.

The arguments developed in the taxonomy section are substantially convincing.

Author Response

Dear reviewer
Thank you so much for taking your time to rewiew our manuscript. The suggestions and revisions you given us are very accurate. The manuscript has been revised using the “Track Changes” function according to your comments and revisions, and the comments have been replied point by point as follows, all revision have been modified in the manuscript.
Sincerely thank again for your review!

Point 1
Location: Table 1.
Comment: Authors should consider to present the pages long Table 1 as Suppl Table.
Response: Thanks very much for your suggestion. Indeed, this table takes up too much space, I have put it in the supplementary materials. The relevant content in the amanuscript is also modified to Table S1.

Point 2
Location:Results, Phylogenetic analysis.
Comment: The authors state that “the trees inferred by ML and BI showed slightly different topologies. (…) the best-scoring ML tree was selected for presentation (Figure 1).” If authors already do the work to analyze their data by two complementary methods, it might be  worth showing both outcomes, i.e. phylogenies, - and if the BI tree were only presented as Suppl Figure.
It is, by the way, not sound to indicate Bayesian posterior probabilities on the branches of the ML tree; please indicate these values in the BI tree.
Response: Thank you very much for your suggestion! I have modified it according to your suggestion. The ML and BI tree are presented separately (because the tree picture is large, I put the BI tree in the supplementary material). The relevant content in the manuscript has also been revised.

Point 3
Location: Discussion, Paragraph 1.
Comment: Authors state that “fungi were clustered into three large clades”:
MLBS and BPP values at the root of the Arthrobotrys and Drechslerella clades (96%/0.94 and 92%/0.94) sufficiently support these clades. This is much less obvious for the Dactylellina clade receiving only 84% MLBS. There is no BPP indicated on this branch. Moreover, the branch rooting a common Arthrobotrys/ Dactylellina clade does not carry any statistical support indications: why? Both MLBS and BPP were below the rspective thresholds? Authors should discuss this outcome and what might be seen as a discrepancy with their interpretation
Response: Thank you so much for your suggestion! It is true that the clustering of genus Dactylellina is really unstable, mainly because the DNA data we used to infer their phylogenetic relationship is not enough. I have revised this section and tweaked the paragraph structure to make the logic smoother. 

Point 4
Location: Phylogenetic analysis and Taxonomy
Comment: The study presents genetic data from 5 pairs of isolates (5 species). Both members of each pair according to descriptions stem from the same geographic origin. These sequences cluster in a pairwise manner with close to 100% MLBS / 1.00 BPP and with (apparently) zero terminal branch length and might therefore represent or 5 geographic clusters or even redundancies of isolation. There is no insight from representing each "new species" by two sequences that are not independent from each other. This appears redundancy of data.
Response: Thank you so much for your suggestion! Yes, each new species has two isolates with essentially identical DNA sequences. Two strains of the same species were isolated from different soil samples (collecting from the same area not far apart). Actually, in earlier versions of the manuscript we only used one strain to report on these five species, but in order to comply with the publication regulations of similar studies in our lab, we deliberately sought additional strains from the our culture collection. For this purpose, the study took a long time to complete the data supplement for additional strains. 

Point 5
Location: Results, Phylogenetic analysis, Paragraph 3.
Comment: The study indicates varying MLBS and BPP values of the clustering of these new sequences with neighboring species. It might be more clearly indicated in the paragraph's argument that these values support assignment of the new fungal isolates at the genus level (i.e., in the manuscript's wording that "five (pairs of) isolates were placed in Arthrobotrys and Drechslerella").
However, the respective values and data can - by their very nature - not lend support to the placement of five new species to these genera. It should be pointed oout at the end of this paragraph that the arguments for species introduction are developed one-by-one and mostly on morphological rather than phylogenetic grounds in the subsequent taxonomy section.
Response: Thanks for your suggestion! Indeed, based on morphology (types of trapping structures), we can pretty much determine which genus these five species belong to. However, phylogenetic studies using multiple genes are needed to determine their specific taxonomic location.

Reviewer 3 Report

The manuscript presents the morphological and phylogenetic characterization of five new fungi that can act as biological control agents for nematodes.

The introduction is well written and describes, with updated literature, the relevance and particularities of the genera Arthrobotrys, Dactyllina and Drechslerella.

The methodology is robust, being widely used in the literature and the results are in accordance with the proposed methodology, being very well presented. The figures and graphics are clear and very well done. I found the “key to known species of Drechslerra” very relevant.

The discussion is well written and, as in the introduction, it presents updated literature and seeks to discuss in depth the data presented in the topic “results”. Given the above and the relevance of the manuscript, I am in favor of its publication.

Author Response

(The authors gave the same response as above.)
